# Nanoemulsion-directed growth of MOFs with versatile architectures for the heterogeneous regeneration of coenzymes

Ke Li[1], Yucheng Zhao[1], Jian Yang[1] & Jinlou Gu [1]✉

As one of the most appealing strategies for the synthesis of nanomaterials with various architectures, emulsion-directed methods have been rarely used to control the structure of metal-organic frameworks (MOFs). Herein, we report a versatile salt-assisted nanoemulsion-guided assembly to achieve continuous architecture transition of hierarchical Zr-based MOFs. The morphology of nanoemulsion can be facilely regulated by tuning the feed ratio of a dual-surfactant and the introduced amount of compatible hydrophobic compounds, which directs the assembly of MOFs with various architectures such as bowl-like mesoporous particle, dendritic nanospheres, walnut-shaped particles, crumpled nanosheets and nano-disks. The developed dendritic nanospheres with highly open and large mesochannels is successfully used as matrix for the co-immobilization of coenzymes and corresponding enzymes to realize the in situ heterogeneous regeneration of $NAD^+$. This strategy is expected to pave a way for exploring sophisticated hierarchical MOFs which can be competent for practical applications with bulk molecules involved.

[1]Key Laboratory for Ultrafine Materials of Ministry of Education, School of Materials Science and Engineering, East China University of Science and Technology, Shanghai 200237, China. ✉email: jinlougu@ecust.edu.cn

The past few decades have witnessed a research boom in metal-organic frameworks (MOFs), which possess intrinsic porous structures through the coordination of organic ligands and metal clusters[1,2]. Such assembly gives MOFs adjustable components and structures in atomic/molecular level[3–5]. On the other hand, the finite topology of secondary building units and short length of organic ligands frequently restrict the structural tunability of MOFs in mesoscale[6,7]. In addition, since their traditional solvothermal synthesis is usually incompatible with the conditions that used for fabricating classical hierarchical materials[8,9], it is thus a challenge in synthetic methodology to delicately control their architectures[10,11]. For example, emulsion-guided growth is one of the most appealing strategies for the synthesis of nanomaterials with versatile architectures but it has rarely introduced to tailor the structure of MOFs[12]. Generally, emulsions are dispersions made up of two immiscible liquid (such as water and oil) combined with surfactants which are responsible to reduce the surface tension. An oil/water (O/W) nanoemulsion system could not only provide the O/W interface as the reaction space but also template the multiple architectures of the materials[13–17]. Therefore, when attempts are made to apply the emulsion-guided method to the synthesis of hierarchical MOFs, two key elements naturally arouse. First, the assembly process needs to be adapted to the change of synthetic solvent from pure organic phase such as dimethylformamide (DMF) to O/W system, and the other is to ensure the effective interaction between the MOFs precursor and emulsion interface.

Actually, improving the synthesis condition of stable MOFs such as UiO-66 family has been a long-pursued goal[18–20]. Our previous works have demonstrated that the addition of Hofmeister ions is favorable to the growth of MOFs in aqueous condition, meanwhile, these ions can mediate the interaction between triblock polymer and MOF precursors[21–23]. On this basis, we expect to develop a nanoemulsion-guided assembly strategy for the fabrication of MOFs with refined hierarchical architectures using triblock polymer as a nanoemulsion stabilizer. Besides the homogeneous growth of the symmetric particles, the introduction of interface by the nanoemulsion should also facilitate the heterogeneous assembly for the asymmetric structure[24], offering great opportunities to obtain MOFs with novel architectures.

In this work, we demonstrate our proposal that the nanoemulsion composed of dual-surfactant (P123 and F127) and a family of hydrophobic aromatic compounds exhibits an excellent versatility as a soft template with the assistance of Hofmeister ion, and allows one to obtain hierarchical Zr-based UiO-66 MOFs with tunable architectures and ultralarge mesopores over 40 nm. The extent of solubilization of aromatic agents such as benzene, toluene, and 1,3,5-trimethylbenzene (TMB) can be systematically varied in micelles of Pluronics according to different proportion of hydrophobic domains[25]. Therefore, by finely tuning the feed ratio of P123 and F127 and the amount of hydrophobic agents, MOFs with various architectures such as bowl-like mesoporous particle, dendritic mesoporous nanospheres, walnut-shaped particles, crumpled nanosheets, and nanodisks can be evolved. Generally, the highly open structure is crucial to the practical applications of MOFs, especially in fields with large molecules involved. Given the stoichiometric consumption and high cost of nicotinamide adenine dinucleotide (NAD$^+$) required by many oxidoreductases, there is an urgent need for the heterogeneous regeneration of NAD$^+$/NADH coenzymes in bio-catalytic industries[26,27]. Therefore, we exemplify that the obtained dendritic mesoporous nanospheres with fully exposed Zr-O sites can afford an ideal platform for anchoring of NAD$^+$ while maintaining its flexibility[28]. Meanwhile, the communication barriers between NAD$^+$ and proximate co-immobilized oxidoreductases can be effectively eliminated, thereby realizing the heterogeneous regeneration of NAD$^+$ redox pair.

## Results

**Preparation and characterization of DMAUiO.** The dendritic mesoporous UiO-66-NH$_2$ (denoted as DMAUiO) nanospheres were first synthesized with P123 and F127 as co-stabilizers, toluene as the oil phase, and ClO$_4^-$ as a mediator[21]. Field-emission scanning electron microscopy (FESEM) image of the formed DMAUiO displays monodispersed spherical morphology with a diameter of around 150 nm (Fig. 1a). The representative transmission electron microscopy (TEM) image further reveals that each DMAUiO possesses radially oriented large channels, which makes the active sites of the whole particle highly accessible from the center to the outer face (Fig. 1b). The opening size of the mesopores is approximately 40 nm, and the thickness of the pore wall is estimated to be about 16 nm. The X-ray diffraction (XRD) pattern of DMAUiO presents characteristic peaks of UiO-66[29], evidencing the formation of crystalline pore walls (Supplementary Fig. 1). The lattice fringes could be observed in the high-resolution TEM (HRTEM) images of DMAUiO (Supplementary Fig. 2), which is consistent with the deduction from the XRD patterns. As revealed by nitrogen sorption analysis, the DMAUiO exhibits a specific surface area of about 1153 m$^2$ g$^{-1}$, and the pore distribution falls in a wide range from 40 nm to macroporous scales due to the packing of nanoparticles (Supplementary Fig. 3). Mercury porosimetry was further applied to determine the meso- and macroporosity of the DMAUiO. As shown in Supplementary Fig. 4, the narrow peak is centered at around 40 nm, which is consistent with the results observed by SEM and TEM.

To get insight into the formation mechanism of such DMAUiO particles, a series of control experiments were conducted and an emulsion-directed assembly pathway was deduced (Fig. 1c). Here, P123 and F127 are expected to stabilize the aromatic swelling agent droplets and guide the self-assembly of MOF precursors with the assistance of ClO$_4^-$. Studies have demonstrated that the solubilization content of swelling agent by Pluronic block copolymers is closely correlated with their hydrophobic PPO domains[25,30]. Specifically, P123 (PEO$_{20}$PPO$_{70}$PEO$_{20}$) with higher fraction of PPO moieties exhibits higher extent of solubilization for toluene than F127 (PEO$_{106}$PPO$_{70}$PEO$_{106}$). Therefore, increasing the ratio of P123 to F127 would favor the combination between toluene and surfactants. When pure F127 or low ratio of P123/F127 were adopted in the synthesis system, the addition of toluene showed negligible effect on the pore size (Supplementary Figs. 5–7). For a certain amount of toluene, the micelles would be swollen and apt to fuse into larger nanoemulsions with the P123/F127 mass ratio increased from 7:8 to 4:1. At the same time, the MOF precursors would nucleate and grow along the extended emulsion interface (Fig. 1c), resulting in the obtained products gradually changing from mesostructure of cage-like pores (Supplementary Fig. 7) to dendritic nanospheres (Fig. 2 and Supplementary Figs. 8 and 9). Meanwhile, with more toluene introduced at fixed P123/F127 ratio, the sizes of nanoemulsions would also increase and further affect the mesostructures (Fig. 2d–l). At P123/F127 mass ratio of 2:1 to 4:1, hierarchical MOFs with hexagonal mesochannels were obtained in the absence of or with a trace amount of toluene (Supplementary Figs. 10 and 11). Further increase the applied toluene, their pore architectures evolved from hexagonal mesochannels to dendritic-like mesopores, and the opening size of the pores could be gradually expanded to around 50 nm (Fig. 2i, l). To further verify the synergistic effects of nanoemulsion and Hofmeister ion, the co-assembly process was also proceeded in the absence of ClO$_4^-$ or surfactants. Upon removing ClO$_4^-$, only particles with sparse pits on the surface could be obtained, indicating that the interaction between surfactants and MOF precursors was too weak to support the guiding effect of the nanoemulsion (Supplementary Fig. 12a).

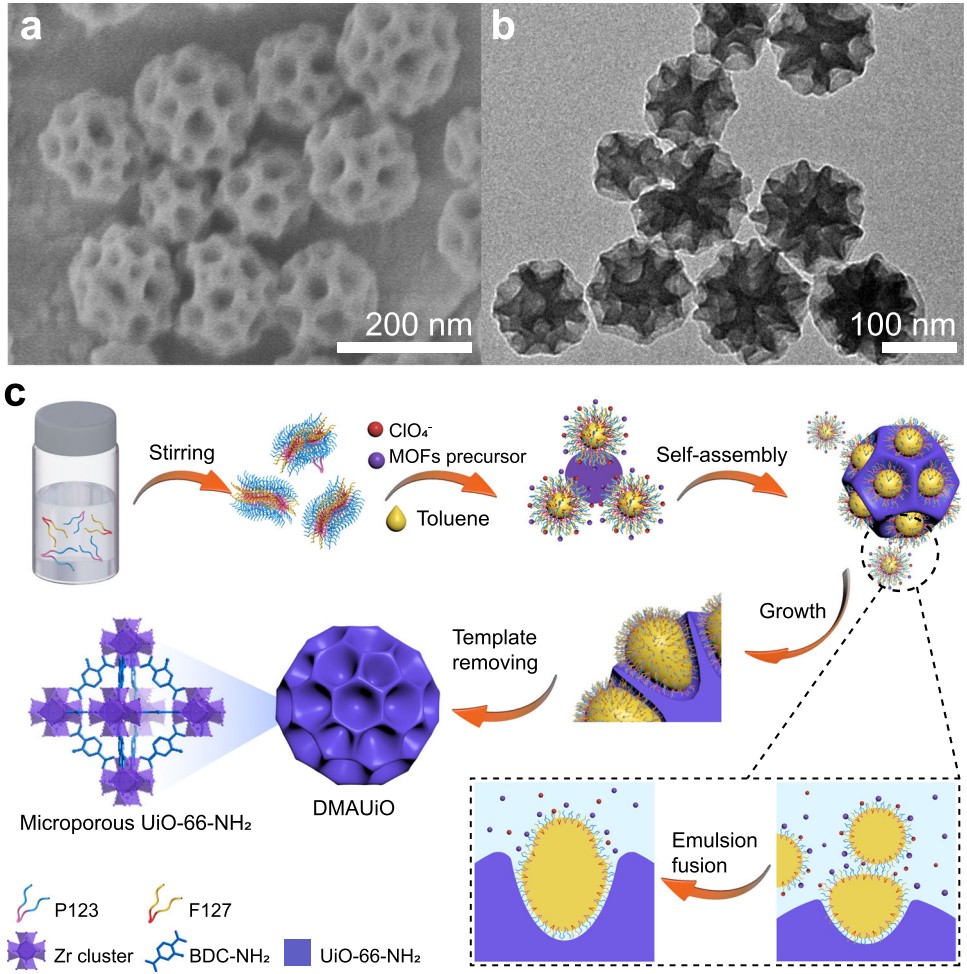

**Fig. 1 Schematic illustration of the formation process of the DMAUiO. a** SEM and **b** TEM images of the DMAUiO synthesized with P123/F127/toluene mass ratio of 1/0.5/1.046. **c** Schematic illustration for the synthesis of the DMAUiO nanospheres with toluene as the oil phase in nanoemulsions.

Meanwhile, stable nanoemulsions could not be formed without the surfactants, thus only non-mesoporous nanoparticles with irregular morphology were formed (Supplementary Fig. 12b, c).

**Preparation of MOFs with various architectures.** Moreover, the number of alkyl substituent of aromatic agents also affects their solubilization content in pertinent surfactants, which further influences the properties of nanoemulsions[25]. With the increase of alkyl-substituent number, the extent of solubilization of benzene, toluene, and TMB in surfactant micelle decreases successively. In view of amphiphilic difference of P123 and F127, one can predict that benzene and toluene would exhibit strong swelling effect on P123, conversely, the addition of TMB in F127 micelles should not lead to appreciable structure transition[25]. Therefore, by altering the type of aromatic agents and precisely controlling the appropriate ratio of two copolymers, the nanoemulsions are expected to exhibit excellent versatility as the soft templates to direct the self-assembly of MOFs. Figure 3 schematically illustrates the structural evolution of the hierarchical MOFs. At P123/F127 mass ratio of 2:1, the excessive TMB leads to the coexistence of small swelled-micelles and large TMB droplets in the system, resulting in the anisotropic growth of asymmetric bowl-like particles with mesopores (Fig. 3a and Supplementary Fig. 13). FESEM image shows that highly open mesochannels expose on the semispherical surfaces of particles (Fig. 4a). TEM image and porosimetry experiments reveal that the pore size is estimated to be around 20–30 nm (Fig. 4f and

Supplementary Figs. 14 and 15). When the mass ratio of P123 to F127 was set between 7:8 and 4:1, as mentioned above, dendritic nanospheres with ultralarge pores would be resulted with toluene as oil phase (Figs. 3b, 4b and 4g). With P123 as a single surfactant, larger but unstable nanoemulsions would be formed. Under the severe agitation, they are prone to deform and merge with each other, which facilitates the formation of walnut-shaped nanoparticles (Fig. 3c)[31]. FESEM image displays that the monodispersed particles are uniform with diameter of around 150 nm, and each particle consists of a continuous bending wall with a thickness of about 15 nm (Fig. 4c). As shown in the TEM image of a single particle, a disordered structure with a wide pore size distribution can be observed (Fig. 4h and Supplementary Figs. 16 and 17). A representative HRTEM image of the particles confirms the crystalline nature of the pore walls (Supplementary Fig. 18). Further increasing the dosage of toluene, the ultralarge nanoemulsions would fuse together (Fig. 3d), thus tending to form crumpled nanosheets with thin walls (Fig. 4d, i and Supplementary Fig. 17). Then, upon replacing the aromatic agent of toluene with benzene, it would cause the strong expansion of the hydrophobic volume (Fig. 3e), hence the morphology of the obtained product transformed to lamellar nanodisks with the diameter of around 100 nm (Fig. 4e, j and Supplementary Fig. 19). The topological structures of all samples were well preserved as verified by XRD (Supplementary Fig. 20), indicating that the introduction of namoemulsions would not deteriorate crystal walls. The architectural variability of the obtained products

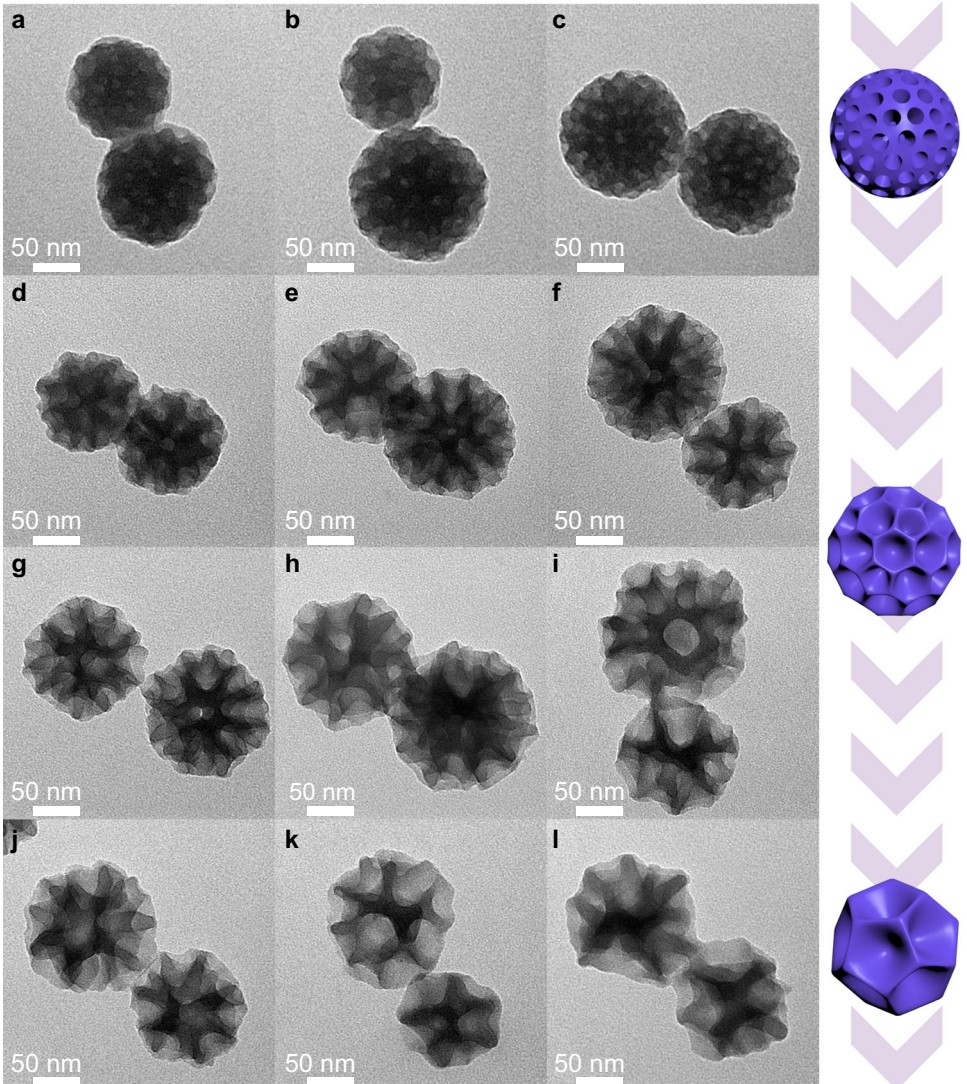

**Fig. 2 TEM images of the DMAUiO synthesized with different P123/F127/toluene ratio.** TEM images of DMAUiO synthesized with P123/F127 mass ratio of 1:2 (**a–c**), 7:8 (**d–f**), 2:1 (**g–i**), 4:1 (**j–l**) and toluene of 60 μL (**a, d, g, j**), 80 μL (**b, e, h, k**), and 100 μL (**c, f, i, l**), respectively. The rightmost panel illustrates the schematic 3D models of the DMAUiO synthesized with different P123/F127/toluene ratio.

further verifies the structure-directing effect of nanoemulsion, opening the possibility to design a more broad range of hierarchical MOFs.

**Deamination of DMAUiO**. Though various architectures of UiO-66-NH$_2$ have been achieved through this nanoemulsion-guided strategy, the formation of parent non-substituted UiO-66 is still a challenge under such mild synthesis condition due to the poor solubility of BDC in aqueous phase. Post-synthetic modification (PSM) is a viable route toward MOFs that are hard to obtain directly[32]. In particular, primary amino groups are regarded as the most versatile sites for PSM, and the reduction of aniline to the corresponding non-substituted aromatic is also a well-known process in organic synthesis[33]. Therefore, using PSM method, the non-substituted dendritic mesoporous UiO-66 (denoted as DMUiO) could be obtained through the deamination of DMAUiO. Here, *tert*-butyl nitrite (*t*-BuONO) and methanol were utilized as the diazotization agent and hydrogen donor to deaminate DMAUiO particles, respectively[34]. The morphology and XRD pattern of the resultant DMUiO resemble to those of parent DMAUiO, evidencing the preserved crystallinity and intact framework (Supplementary Fig. 21a, b). The characteristic UV-Vis absorption of BDC-NH$_2$ in

the region of ~300–400 nm disappears after deamination (Supplementary Fig. 21c). The $^1$H nuclear magnetic resonance ($^1$H NMR) spectrum of the digested DMUiO displays a typical singlet of terephthalic acid at 7.7 ppm, and no signals of BDC-NH$_2$ could be detected, further demonstrating the complete deamination of the DMAUiO (Supplementary Fig. 21d).

**Heterogeneous regeneration of the coenzymes in DMUiO**. Oxidoreductases are widely used in bio-catalytic industries, and generally require a coenzyme such as NAD(H) to work[35,36]. Unlike enzymes, coenzymes usually undergo stoichiometric transformation with the substrates in a chemical reaction. Therefore, the efficient regeneration and recyclability of these expensive coenzymes are of great significance. Given the abundant exposed Zr-O active sites and ultralarge pores, DMUiO is expected to be an ideal matrix to balance the retention and reaction activity of NAD$^+$. By forming Zr-O-P bonds[37], NAD$^+$ can be stably suspended on the pore walls, and it could still be mobile locally thanks to the sufficient surrounding free space, ensuring its binding affinity to the active pocket sites of oxidoreductases. Meanwhile, the highly open structure and large pore space can realize the co-immobilization of desired enzymes[38–43],

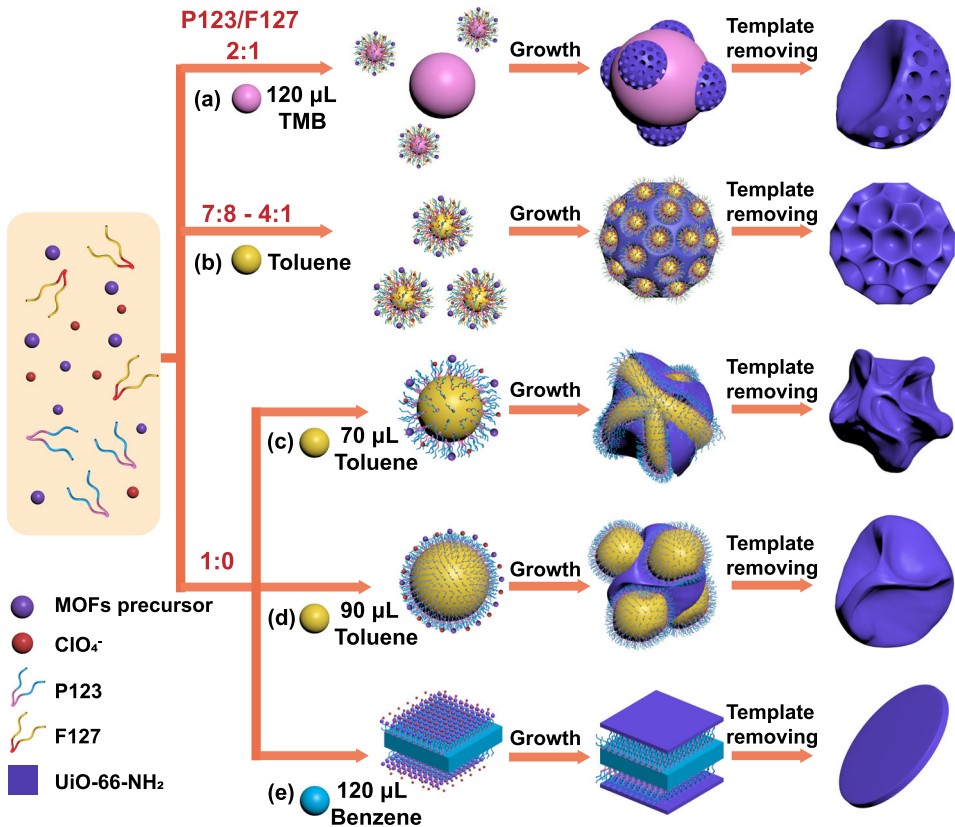

**Fig. 3 Schematic illustration for the formation process of hierarchical Zr-based MOFs with various architectures using the proposed nanoemulsion-guided assembly strategy.** The formation process of **a** bowl-like, **b** dendritic, **c** walnut-shaped, **d** crumpled nanosheet, and **e** nanodisk UiO-66-NH$_2$.

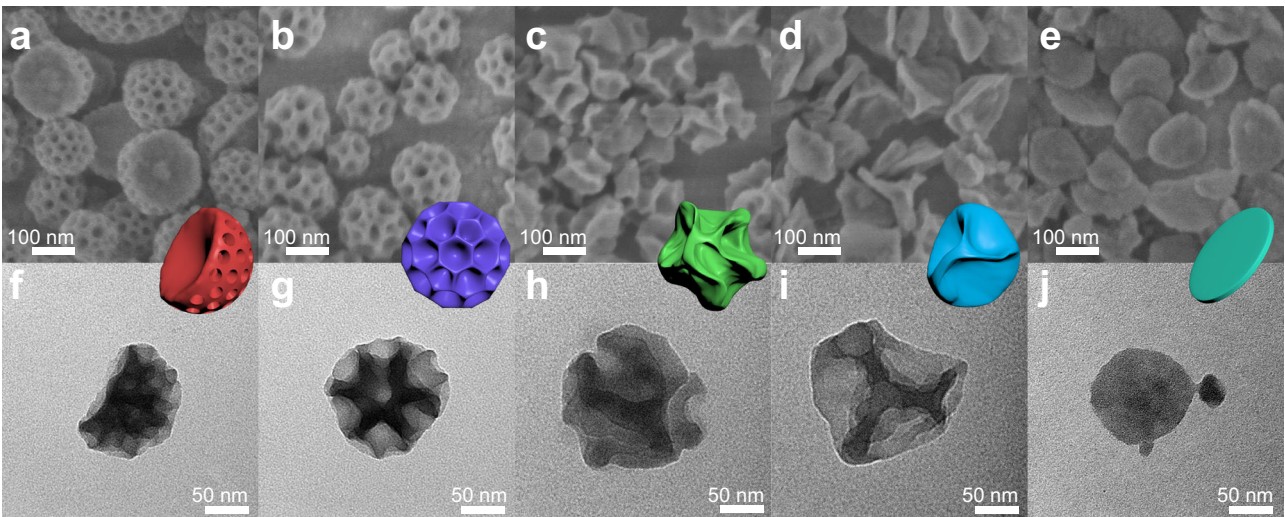

**Fig. 4 Structural characterizations of the hierarchical Zr-based MOFs.** SEM (**a**–**e**) and TEM (**f**, **g**) images of hierarchical Zr-based MOFs synthesized with different P123/F127 mass ratio and swelling agent: **a**, **f** 2:1, 120 μL TMB; **b**, **g** 2:1, 80 μL toluene; **c**, **h** 1:0, 70 μL toluene; **d**, **i** 1:0, 90 μL toluene; **e**, **j** 1:0, 120 μL benzene. The insets in Fig. 4 show the 3D model of the corresponding architectures.

minimize the mass transfer resistance and facilitate the inter-communication between coenzymes and enzymes.

As a proof of concept, alcohol dehydrogenase (ADH) and NAD$^+$ were first co-immobilized in DMUiO with open pore sizes of 20 and 40 nm (denoted as DMUiO-20 and DMUiO-40). As quantitatively determined from UV-Vis spectra, the amounts of encapsulated NAD$^+$ and ADH in DMUiO-40 are estimated to be about 280 and

177 μg mg$^{-1}$, respectively, whereas those in DMUiO-20 are only about 192 and 113 μg mg$^{-1}$. In this system, ethanol can be converted to acetaldehyde catalyzed by ADH with concomitant reduction of NAD$^+$ to NADH (Fig. 5a(I)). The reaction process was evaluated by monitoring the blue fluorescence of NADH emitted at around 440 nm[44]. Figure 5b shows a time-dependent emission increase in the NAD$^+$@ADH@DMUiO-40 system. Then, the NAD

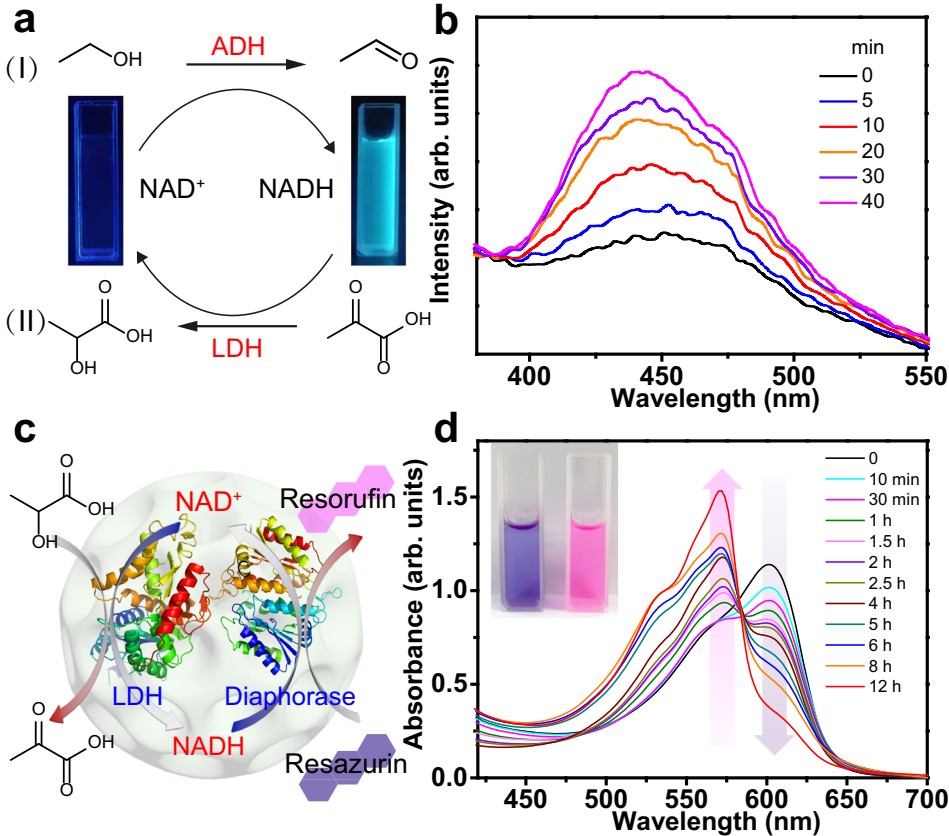

**Fig. 5 Heterogeneous regeneration of the coenzymes in DMUiO. a** Scheme of the regeneration of NAD$^+$/NADH catalyzed by ADH and LDH. The insets are digital pictures of the NAD$^+$ (left) and NADH (right) solutions under UV light. **b** Fluorescence spectra of the NAD$^+$@ADH@DMUiO-40 system after the addition of ethanol. **c** Scheme of the regeneration of NAD$^+$/NADH catalyzed by LDH and diaphorase. **d** UV-Vis absorption of the LDH@diaphorase@NAD$^+$@DMUiO system after the addition of L-lactic acid at different time. The inset in (**d**) shows the digital picture to directly illustrate color change of the reaction solution.

$^+$ reduction in DMUiO-20 and DMUiO-40 was compared with free NAD$^+$/ADH system in aqueous solution (Supplementary Fig. 22). The reaction profile for the 20 min indicates that the activity of free NAD$^+$/ADH system was lower than those of NAD$^+$@ADH@DMUiO ones. Meanwhile, due to the relatively large pore size and highly open structure of DMUiO-40, NAD$^+$@ADH@DMUiO-40 system exhibited faster reaction rate than that of NAD$^+$@ADH@DMUiO-20.

To verify the reusability and regeneration of NAD$^+$, the NAD$^+$@ADH@DMUiO-40 system was coupled with lactate dehydrogenase (LDH), which can catalyze the reversible reaction between pyruvic acid and L-lactic acid with the conversion of NAD$^+$/NADH (Fig. 5a(II)). After the oxidation of ethanol, the NADH@ADH@LDH@DMUiO composite was separated from the system, and pyruvic acid was added to realize the regeneration of NAD$^+$. The fluorescence emission of NADH was recorded and shown in Supplementary Fig. 23. The fluorescence intensity decreases gradually as time prolongs, indicating the oxidation of NADH to the non-fluorescent NAD$^+$, which confirmed that NAD$^+$ can be recovered from solution and regenerated in the presence of LDH.

To more directly illustrate the regeneration cycle of coenzymes, LDH, diaphorase, and NAD$^+$ were co-immobilized in DMUiO-40 to construct a cascade catalytic system (Fig. 5c). In the first step, LDH catalyzes the conversion of L-lactic acid to pyruvic acid with the concomitant reduction of NAD$^+$ to NADH; in the second half cycle, a fluorescent dye of resazurin was used as a reducing indicator[45]. In the presence of diaphorase, the blue resazurin can be reduced to the pink resorufin, meanwhile, the

resulting NADH can be oxidized back to the NAD$^+$. As shown in Fig. 5d, the intensity of the absorption bands of resazurin at 600 nm decreased gradually, and the bands at 571 nm appeared with enhanced intensity, corresponding to the absorption of resorufin. Meanwhile, the color change from blue to pink could be observed by naked eyes. Control experiments demonstrated that the reaction activity of free LDH/diaphorase/NAD$^+$ system was lower than that of co-immobilized system (Supplementary Fig. 24), agreeing with our proposal that DMUiO particles could provide an effective platform for the heterogeneous multi-enzymatic regeneration of NAD$^+$.

## Discussion

In summary, a salt-assisted nanoemulsion-guided assembly has been demonstrated to construct hierarchical Zr-based MOFs. By delicately adjusting the feed ratio of P123/F127 and introduced amounts of hydrophobic aromatic compounds with different alkyl substitution, the production of MOFs with various architectures is realized. The highly accessible active sites together with the large and open spaces in such architectures make them competent in a broad range of applications with bulk molecules involved. The obtained DMUiO with tunable pore size and highly open channels from center to surface was exemplified to provide an ideal platform for the coenzyme-dependent cascade enzymatic reaction, realizing the effective heterogeneous regeneration of expensive NAD$^+$. We expect that the present strategy may offer a route toward the rational design of hierarchically MOFs with novel architectures and broaden the application potentials of MOFs with biomacromolecules involved.

## Methods

**Synthesis of dendritic mesoporous UiO-66-NH₂ (DMAUiO)**. In a typical synthesis, 50 mg P123 and 25 mg F127 were dissolved in 3 mL deionized water, and then AA (0.4 mL), $NaClO_4·H_2O$ (150 mg, 1 mmol) and 80 μL toluene were added. The mixture was stirred to form a nanoemulsion solution. Subsequently, ZrO(-NO₃)₂·2H₂O (115.6 mg, 0.5 mmol) and BDC-NH₂ (50 mg, 0.28 mmol) were added into the above mixture. Then, the mixture was stirred for 12 h at 40 °C. The resultant solid was isolated by centrifugation and washed with water once and DMF twice. To remove the template, the as-synthesized sample was soaked in ethanol for two days at 60 °C, during which time the ethanol was renewed every day. Finally, the product was dried overnight at 60 °C under vacuum.

MOFs with other architectures were synthesized using otherwise the same procedure except the amounts of reagents were different as listed in Supplementary Table 1.

**Synthesis of dendritic mesoporous UiO-66 (DMUiO)**. 10 mg DMAUiO was mixed with 1 mL t-BuONO and placed at −20 °C for 1 h. Next, 1 mL MeOH was added into the above mixture and placed at −20 °C for another 1 h. Then the reaction mixture was heated at 80 °C for 18 h. Finally, the excessive t-BuONO was washed out with MeOH and dried overnight at 60 °C under vacuum.

## Data availability

The data generated in this study are provided in the Supplementary Information/Source data file. Source data are provided with this paper.

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

## Acknowledgements

This work was financially supported by the Natural Science Foundation of China (J.G., 21975072; J.Y., 51902106), the Natural Science Foundation of Shanghai (J.G., 18ZR1408700).

## Author contributions

K.L. and J.G. designed the research. K.L. and Y.Z. performed the experiments and characterizations. K.L., J.Y., and J.G. co-wrote the manuscript.

## Competing interests

The authors declare no competing interests.
