## [Peer Review File · Nature Communications]

Nanoemulsion-directed growth of MOFs with versatile architectures for the heterogeneous regeneration of coenzymesREVIEWER COMMENTS

Reviewer #1 (Remarks to the Author):

The manuscript reports on an approach to control morphology of MOF particles using nanoemulsion templates of tunable properties. The approach appears highly successful and is likely to be applicable to other MOF compositions. The work is carefully executed. Still, it is recommended to directly show that the particles of controlled morphology have crystalline walls, which may be achieved via high-magnification TEM or electron diffraction for instance. The combination of well-resolved XRD patterns with high mesopore volume inferred from nitrogen adsorption leaves little doubt that the claims of the authors are correct, but the direct TEM or electron diffraction evidence would be welcome. If the authors can address the above issue, the manuscript will be fully recommendable for publication. The TEM imaging at high magnification, if achieved, may also allow to provide the understanding of the interface between the nanoemulsion phase and the MOF phase. Also, the authors may want to compare and contrast the structures they document with the structures of other compositions reported in the literature using similar surfactant-mixture templates, including Ref. [28] and Chem. Mater. 2017, 29, 4675.

Reviewer #2 (Remarks to the Author):

This manuscript describes the preparation of new and exciting morphologic architectures of NH₂-UiO-66(Zr) MOF material (which can be PSM converted into non-functionalized UiO-66(Zr)) by combining surfactants, Hofmeister ions and biphasic media. In addition, the generated meso- and/or macropores has been successfully used as support for the regeneration of the coenzyme NAD⁺ through a cascade reaction. This is a very original, well-designed and well-written work and their results are of high interest for the community working in MOFs as supports of heterogeneous catalysts such as enzymes. That is why I consider that this manuscript is potentially publishable in Nature Communications. Before, it is accepted I would like the authors consider the following comments:

1. It should be clarified if the Pore Size Distribution curves has been generated from adsorption or from desorption branches.
2. The pores/cavities generated by the surfactant-Hofmeister-biphasic media are quite large (sometimes away from the mesopore range according to IUPAC). Moreover, some of the samples contain pores of different sizes within the same particles (see for instance the bowl-like particles). In my opinion, PSD from nitrogen isotherms could not be the best technique for this goal. As a consequence, the meso-macropores easily seen by SEM/TEM, which are of relatively narrow size distribution, could not be identified clearly in the presented PSD. In principle, mercury porosimetry is more adequate. I recommend the authors to study these samples (at least, some of them) by means of such technique.
3. The experimental part of the sample digestion of the MOF as a previous step of NMR characterization should be given in the experimental part.

Response to the Comments

Reviewer 1

General Comments

The manuscript reports on an approach to control morphology of MOF particles using nanoemulsion templates of tunable properties. The approach appears highly successful and is likely to be applicable to other MOF compositions. The work is carefully executed.

Response

Many thanks for the positive comments, constructive suggestions and kind recommendation from the reviewer.

We have carefully checked and revised the original manuscript and taken reviewer's comments into full consideration. The following is our full response to the comments point by point.

Specific points

1) Still, it is recommended to directly show that the particles of controlled morphology have crystalline walls, which may be achieved via high-magnification TEM or electron diffraction for instance. The combination of well-resolved XRD patterns with high mesopore volume inferred from nitrogen adsorption leaves little doubt that the claims of the authors are correct, but the direct TEM or electron diffraction evidence would be welcome. If the authors can address the above issue, the manuscript will be fully recommendable for publication. The TEM imaging at high magnification, if achieved, may also allow to provide the understanding of the interface between the nanoemulsion phase and the MOF phase.

Response

Many thanks for the constructive suggestion from the reviewer.

We fully agree with the reviewer that HRTEM could more directly demonstrate the crystalline structure of the walls, and we have tried to perform HRTEM characterization as the reviewer suggested. As shown in Supplementary Figs 2 and 18, the lattice fringes could be observed in the HRTEM images of dendritic and walnut-shaped UiO-66-NH₂, demonstrating that the walls are crystallized and in good agreement with the deduction from the XRD patterns. However, due to the extreme instability of MOFs upon high-energy electron beam irradiation (*J. Am. Chem. Soc.* 2020, 142, 17224; *Nature Materials*, 2017, 16, 532; *CrystEngComm*,

2013, 15, 9356), despite our efforts for several times, only limited resolution images could be obtained. We are very sorry that we cannot provide HRTEM images with more structure details, and we sincerely hope that the reviewer can understand this situation.

We have added the HRTEM results in the supplementary information (Please see page 4, SI pages 4 and 11) as the reviewer suggested.

Supplementary Figure 2. (a) A typical HRTEM image of DMAUiO. (b) Enlarged image of the highlighted area shown in (a).

Supplementary Figure 18. (a) A typical HRTEM image of walnut-shaped UiO-66-NH₂. (b) Enlarged image of the highlighted area shown in (a).

2) Also, the authors may want to compare and contrast the structures they document with the structures of other compositions reported in the literature using similar surfactant-mixture templates, including Ref. [28] and Chem. Mater. 2017, 29, 4675.

Response

Lou and Lruk groups has made great effort to construct meso- and macroporous structures with carbon (*Angew. Chem.* 2018, 130, 6284) and silica (*Chem. Mater.* 2017, 29, 4675) as matrices using similar surfactant-mixture templates. Compare to these amorphous compositions, as confirmed by XRD and HRTEM, the pore walls of our materials in this work are intrinsically porous at the microporous length scale due to their defined frameworks. Therefore, the resulting material possesses hierarchical micro- and mesostructure, and enable the fast diffusion of substances within it, making it an ideal platform for mass transport. Meanwhile, MOFs allow for precise design of framework structures and fine tailoring of pore environments at the molecular or atomic level. Therefore, MOFs can be extended to several analogous frameworks that are synthesized from different metallic components and identical linkers, and works in this direction is in progress.

We have added the important reference (*Chem. Mater.* 2017, 29, 4675) into the corrected manuscript.

Reviewer 2

General Comments

This manuscript describes the preparation of new and exciting morphologic architectures of NH₂-UiO-66(Zr) MOF material (which can be PSM converted into non-functionalized UiO-66(Zr)) by combining surfactants, Hofmeister ions and biphasic media. In addition, the generated meso- and/or macropores has been successfully used as support for the regeneration of the coenzyme NAD⁺ through a cascade reaction. This is a very original, well-designed and well-written work and their results are of high interest for the community working in MOFs as supports of heterogeneous catalysts such as enzymes. That is why I consider that this manuscript is potentially publishable in Nature Communications. Before, it is accepted I would like the authors consider the following comments

Response

Thanks for the precise summary, constructive comments and kind recommendation from the reviewer. We have carefully checked and revised the original manuscript according to the comments from the reviewer. The following is our full response to the comments point by point.

Specific points

1) It should be clarified if the Pore Size Distribution curves has been generated from adsorption or from desorption branches.

Response

Many thanks for the constructive suggestion from the reviewer.

The pore size distribution curves were generated from adsorption branches. We have added the above description in the supplementary information as the reviewer suggested (Please see SI page 1).

2) The pores/cavities generated by the surfactant-Hofmeister-biphasic media are quite large (sometimes away from the mesopore range according to IUPAC). Moreover, some of the samples contain pores of different sizes within the same particles (see for instance the bowl-like particles). In my opinion, PSD from nitrogen isotherms could not be the best technique for this goal. As a consequence, the meso-macropores easily seen by SEM/TEM, which are of relatively narrow size distribution, could not be identified clearly in the presented PSD. In principle, mercury porosimetry is more adequate. I recommend the authors to study these samples (at least, some of them) by means of such technique.

Response

Thanks for the constructive suggestion from the reviewer.

In order to complement the evaluation of porosity in samples by N₂ sorption, as suggested by the reviewer, the mesoporosity of the bowl-like, dendritic and walnut-shaped UiO-66-NH₂ were determined by the mercury porosimetry method. For the DMAUiO, the narrow peak is centered at around 40 nm, which is consistent with the results observed by SEM and TEM (Supplementary Fig. 4). As shown in Supplementary Fig. 15, the presence of large mesopores with pore size of about 32 nm in bowl-like UiO-66-NH₂ is obvious, corresponding to the rise in the cumulative intrusion curve (Red curve in Supplementary Fig. 15). Though the bowl-like particles contain both meso- and macropores, the curve still shows unimodal distribution. This is reasonable since the shallow macropores exist almost in a plane form which is undetectable by mercury porosimetry. The broad and draggy macropores are derived from inter-particle packing due to the small particle size. As shown in Supplementary Fig. 16, walnut-shaped UiO-66-NH₂ exhibits a pore distribution centered at around 34 nm, confirming that the larger mesopores are open at the surface of the particle.

We have added the mercury porosimetry results in the supplementary information as the reviewer suggested (Please see page 4, SI pages 5 and 10).

Supplementary Figure 4. Cumulative pore volume curve (red) and pore size distribution (black) derived from mercury porosimetry analysis for the sample of DMAUiO synthesized with 80 μL toluene when the P123/F127 feed ratio is 2:1.

Supplementary Figure 15. Cumulative pore volume curve (red) and pore size distribution (black) for the sample of bowl-like UiO-66-NH₂ by mercury porosimetry analysis.

Supplementary Figure 16. Cumulative pore volume curve (red) and pore size distribution (black) for the sample of walnut-shaped UiO-66-NH₂ by mercury porosimetry analysis

3) The experimental part of the sample digestion of the MOF as a previous step of NMR characterization should be given in the experimental part.

Response

Thanks for the kind suggestion from the reviewer.

We have added the digestion step of NMR characterization in the supplementary information as the reviewer suggested. The following description was added to SI page 2:

Prior to ¹H NMR measurement, samples were digested at 80 °C for 2 h by adding 600 μL of a 1M NaOH in D₂O solution to a centrifuge tube containing 20 mg of sample. After the digestion, the inorganic component was removed by centrifugation and the supernatant was transferred into an NMR tube.

REVIEWER COMMENTS

Reviewer #1 (Remarks to the Author):

The authors have responded to the reviewer's comments and revised the manuscript accordingly, thus satisfactorily addressing the reviewer's comments. The manuscript is now recommended for publication in Nature Communications.

Reviewer #2 (Remarks to the Author):

In my opinion, the first version of this manuscript was already quite good. Furthermore, this version, which has been modified following the advices given by Reviewers, is even better. I am happy with the modifications made by the authors. The only controversial point could be the demonstration that the walls are crystalline in a more visual way by means of HRTEM. Unfortunately, the shown TEM micrographs are not entirely conclusive in that aspect. Nevertheless, I am aware of the extreme difficulty of observing the crystallographic planes in the walls of MOFs, due to their instability to the beam, especially for these materials having nano-walls. The only way is to use low and controlled incident beam doses but even though it is not an easy task (see for instance, ChemCatChem 2015, 7, 3719 – 3724).

I think the manuscript is ready to be published in Nature Communication.

Response to the Comments

Reviewer 1

General Comments

The authors have responded to the reviewer's comments and revised the manuscript accordingly, thus satisfactorily addressing the reviewer's comments. The manuscript is now recommended for publication in Nature Communications.

Response

Many thanks for the kind recommendation from the reviewer.

Reviewer 2

General Comments

In my opinion, the first version of this manuscript was already quite good. Furthermore, this version, which has been modified following the advices given by Reviewers, is even better. I am happy with the modifications made by the authors. The only controversial point could be the demonstration that the walls are crystalline in a more visual way by means of HRTEM. Unfortunately, the shown TEM micrographs are not entirely conclusive in that aspect. Nevertheless, I am aware of the extreme difficulty of observing the crystallographic planes in the walls of MOFs, due to their instability to the beam, especially for these materials having nano-walls. The only way is to use low and controlled incident beam doses but even though it is not an easy task (see for instance, ChemCatChem 2015, 7, 3719 – 3724).

I think the manuscript is ready to be published in Nature Communication.

Response

Many thanks for the positive comments and the kind recommendation from the reviewer. We are also very grateful to the reviewer for understanding the difficulties we encountered during the HRTEM measurement.